# Estimating firms' emissions from asset level data helps revealing (mis)alignment to net zero targets

Hamada Saleh [1], Stefano Battiston [2,3], Irene Monasterolo [4,5,6], Thibaud Barreau [1] & Peter Tankov [1,7] ✉

We develop a bottom-up methodology to estimate companies' (mis)alignment to net-zero scenarios. The approach relies on asset-level data for individual production units, enabling a detailed estimation of corporate emissions trajectories. We apply the methodology to the steel sector globally and find that companies' projected emissions for 2030 exceed those implied by the International Energy Agency's (IEA) Net Zero Emissions (NZE) scenario by between 10% and 22%, depending on the assumptions about the future evolution of emission factors of steel production. Further, we find that projected emissions for 2030 exceed companies' aggregate stated targets, even under the optimistic assumption of electricity supply decarbonization rate following the net-zero scenario, with the gap primarily driven by the largest steel companies. Our results show that a bottom-up asset-level approach allows for a reality check of companies' contributions to national decarbonization plans. This, in turn, is crucial to inform more targeted industrial policies for decarbonization, and regulatory disclosure.

Avoiding the most adverse impacts of climate change requires achieving net-zero greenhouse gas (GHG) emissions at the global level before 2050[1]. In this context, the question of reliably estimating companies' alignment to net-zero GHG emission scenarios (hereafter, net-zero alignment) is crucial both to inform climate policies and regulations and to enable global financial institutions to evaluate and manage climate-related risks.

On the one hand, earlier works have introduced methodologies to set net-zero targets and assess firms' net-zero alignment[2–4]. These also include the Science-Based Targets Initiative[5]. On the other hand, other works have focused on assessing the alignment of financial portfolios to decarbonization targets, such as 2 °C or net-zero 2050[6–8], using as input estimates of companies' projected emissions. However, these estimates are most often based on firms' self-reported GHG emissions and are obtained by extrapolating historical GHG emissions using a linear regression.

Linear extrapolation from historical emissions suffers from three main shortcomings. First, often only few (less than 5) years of data on emissions are available: for example, 20,202 companies disclosed their climate performance to CDP in 2023, but only 9526 did so in 2020 (see https://cdp.net/en/companies/cdp-2023-disclosure-data-factsheet). The emission trajectories can be volatile due to exogenous shocks (e.g., the COVID-19 pandemic and the war in Ukraine). In this context, the use of machine learning techniques to fill the data gaps or to reduce noise is of limited help. Indeed, machine learning methods for estimating past emissions for non-reporting companies[9–11] are trained on emission data from reporting companies, which may be subject to biases and inconsistencies[12,13].

Second, even if longer time series are available (e.g., 10 years), linear extrapolation can lead to either overestimate or underestimate future emissions. For instance, if a company has been on an emission growth trajectory in the last 5 years but is now in the course of a

[1]Institut Louis Bachelier, Paris, France. [2]University of Zurich, Zurich, Switzerland. [3]University of Venice, Venice, Italy. [4]University of Utrecht, Utrecht, Netherlands. [5]CEPR, Paris, France. [6]WU Wien, Vienna, Austria. [7]CREST, ENSAE, Institut Polytechnique de Paris, Palaiseau, France. ✉e-mail: peter.tankov@ensae.fr

transformation to low-carbon technologies, projections for 2030 will exceed the actual emissions levels. Conversely, if the company has been decreasing emissions due to exogenous shocks, but it has not invested in low-carbon technologies, the projected emissions for 2030 are lower than today, while the actual ones can be much higher.

Third, the above problems are exacerbated by the use of metrics of emissions' financial intensity, such as the ratio of emissions to revenues or the ratio of emissions to market capitalization, which are more volatile than emission measures expressed in physical units.

Thus, while using companies' self-reported emissions to estimate future emission trends has been an important development for disclosure in recent years, from a policy and investment perspective, there is a need to independently validate such self-reported estimates. In particular, the misestimation of emissions is problematic because decision makers within the companies themselves or in public authorities may find out too late that companies' stated targets are not credible, or will not be met. When this occurs systematically across companies and sectors, it can jeopardize the achievement of the 2 °C objective. If, instead, discrepancies can be detected early on, countermeasures can be put in place, including a revision of disclosure requirements and of the reporting regulation.

To address these problems here, we develop a methodology to estimate companies' alignment with net-zero scenarios. Our approach makes use of asset-level data, i.e., data on companies' emissions at the level of their production units. Instead of relying on linear extrapolations of emissions, we take into account the opening/closing dates of each production unit, as well as its technology and its emission physical intensity (i.e., tonnes of $CO_2$e per tonnes of physical output). Further, for technologies that use electricity as an input, we consider alternative scenarios for the decarbonization rate of the electricity supply. By aggregating emissions of production units at the level of individual companies (hereafter, bottom-up approach), the methodology yields an independent estimate of future emissions trajectories that reflect the objective capabilities of companies in terms of technologies and investments on the ground. These estimates can then be compared to those provided, at the sectoral level, by the reference scenarios of the International Energy Agency (IEA), or to the decarbonization targets stated by companies themselves, when available.

In this work, we apply the methodology to the steel sector worldwide. Steel is a key sector for decarbonization scenarios[14–17], because it will remain an essential input material for many economic activities, and it is one of the metals required for expanding electrification infrastructures (e.g., railways, power transmission, renewable energy facilities).

We find that our bottom-up projected emissions in 2030 exceed the emissions trajectories in the net-zero scenario of the IEA to an extent varying between 10% and 22%, depending on the assumptions on the rate of decarbonization of the electricity supply to the steel sector and on the future evolution of the emission factors of steel production. Further, the sum of projected emissions in 2030 across companies exceeds the sum of emissions foreseen by companies' stated targets, even in the optimistic case that the decarbonization rate of electricity supply follows the net-zero scenario. Notably, this discrepancy is driven by the largest companies.

Our methodology could be applied to other sectors as well, in particular those for which production is carried out by means of a small set of alternative technologies for which the emission physical intensity is known. In Supplementary Section 3 (Cross-sector applicability) and Supplementary Table 3 (Plant-level and technology data availability for various sectors), we discuss in detail the challenges for applying it to cement, aluminum, pulp & paper and power sectors, and point the reader to relevant data sources.

While ongoing developments in Artificial Intelligence can further facilitate the automatic extraction of emission data from firms' reports, the need will remain for complementary and independent estimates of companies' future emissions trajectories. Our methodology can contribute to improving the quality of emissions disclosure and standards for firms' emissions on alignment plans. This information, in turn, is crucial to support credible climate policies and effective international supervision.

## Results

### Methodology for projecting future company emissions from asset-level data

Our first contribution is a bottom-up methodology for estimating future carbon emissions of companies based on their assets (i.e., production units) and based on climate policy scenarios. The workflow of the methodology is illustrated in Fig. 1 and is described in detail in section "Methodology".

The methodology is based on four types of data sources:

i. Scenario data. This includes projections of production volume in relevant climate scenarios and projections of carbon intensity of electricity generation in the same scenarios (for estimating Scope 2 emissions). In our application to the iron and steel sector, we use the IEA World Energy Outlook 2023 scenarios, hereafter referred to as IEA scenarios.

ii. Asset-level data. This is the main source of data for the methodology, which includes, for each company, capacity, technology, and past production volume of each of their assets (production units), as well as their planned opening/closing dates (when available).

iii. Technology-specific data. This includes past and present carbon emission factors per physical unit of production for each relevant production technology at the available granularity (world or country-level), as well as the information about potential future reductions of these emission factors (best available technologies to reduce emissions, their abatement potential and present state of deployment).

iv. Reported emission data. This includes company-level reported emissions from CDP (formerly Carbon Disclosure Project) and Thomson Reuters (Refinitiv).

Scenario, asset-level and technology data are then combined to estimate the future raw bottom-up emissions. This step involves: (a) the estimation of the future utilization rates based on past production and scenario-based future production; (b) the estimation of the future emission factors based on technology data and scenario-based electricity generation carbon intensity; (c) the identification of which assets will be in operation at a given future date, based on information on opening/closing dates.

The estimation of future raw bottom-up emissions is then aggregated at the firm level and translated into future emissions by means of a statistical model (linear regression), trained on past bottom-up emissions aggregated at the firm level, and past firms' reported scope 1 and 2 emissions. This step enables us to account for emission sources not directly related to the production process, such as electricity consumption for uses other than electric arc furnace steel production.

To quantify the future utilization rate of steel plants, we consider two alternative plausible assumptions about their evolution. In the first method (the country-level utilization rate), all steel plants of a given country are assumed to have the same utilization rate, determined by taking the past utilization rate and adjusting it with a common global multiplier to match the scenario-based production. In the second method (constant market share), all companies are assumed to have a constant market share, and all plants of a given company are assumed to have the same utilization rate. Despite being based on different assumptions, the two methods give very similar results. For this reason, in the discussion of the main results of the paper, we only use the constant market share method. In the Methods section and in Supplementary Fig. 5, we present the results for the country-level

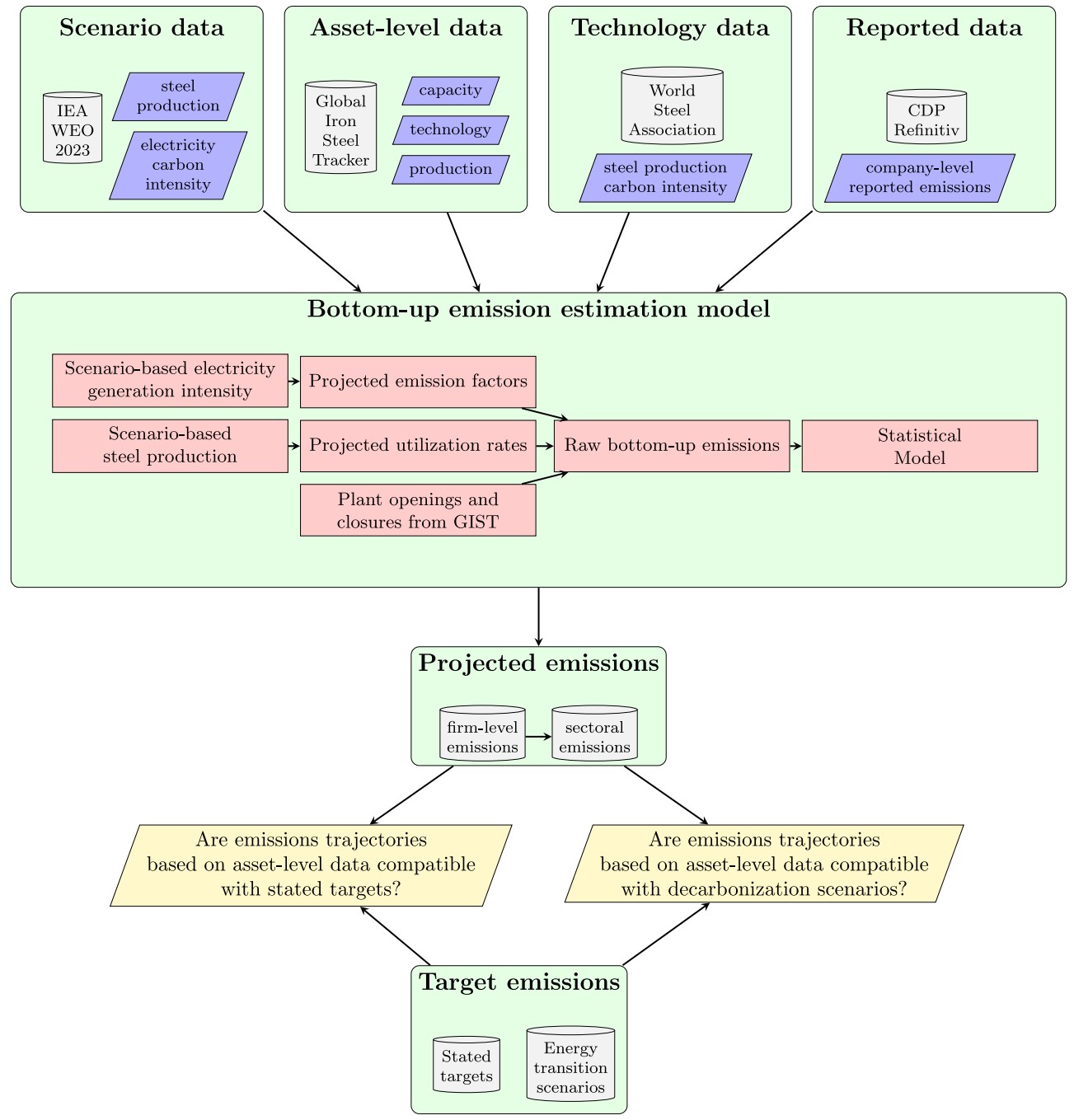

**Fig. 1 | Illustration of the bottom-up alignment assessment framework, applied here to the Iron & Steel sector.** From top to bottom. The model uses as inputs data on scenarios, productive assets, technology and reported emissions. This information feeds a bottom-up emissions estimation model that is able to reproduce firms' past emissions (back-testing) and to produce future estimations conditioned to climate policy scenarios for the sector (forecasting). The resulting firms' estimated emissions are compared with firms' stated alignment targets and with the emissions path of decarbonization scenarios (e.g., the Net Zero 2050 scenario from the International Energy Agency).

utilization rate method, and also, to explore the uncertainty associated with projecting future plant utilization rates, we compute upper and lower bounds on future emissions by affecting all scenario-based production to, respectively, the most polluting and the least polluting plants of each country.

The future firm-level emissions based on asset-level data are then used to answer two questions related to net-zero alignment of the iron and steel sector:

- To what extent is the iron and steel sector as a whole aligned with the net-zero scenario? To answer this question, we aggregate our estimated firm-level future emission trajectories at the sector level and compare them with the IEA Net Zero 2050 (NZE) scenario for the steel sector. In Supplementary Figs. 1 and 2, we also compare the estimated firm-level future emission trajectories with IEA Announced Pledges (APS) and Stated Policies (STEPS) scenarios.
- Are the future decarbonization trajectories of the companies in the iron and steel sector, as estimated from asset-level data, consistent with the stated decarbonization objectives of the same companies? To answer this question, we aggregate our estimated firm-level future emission trajectories only for companies for

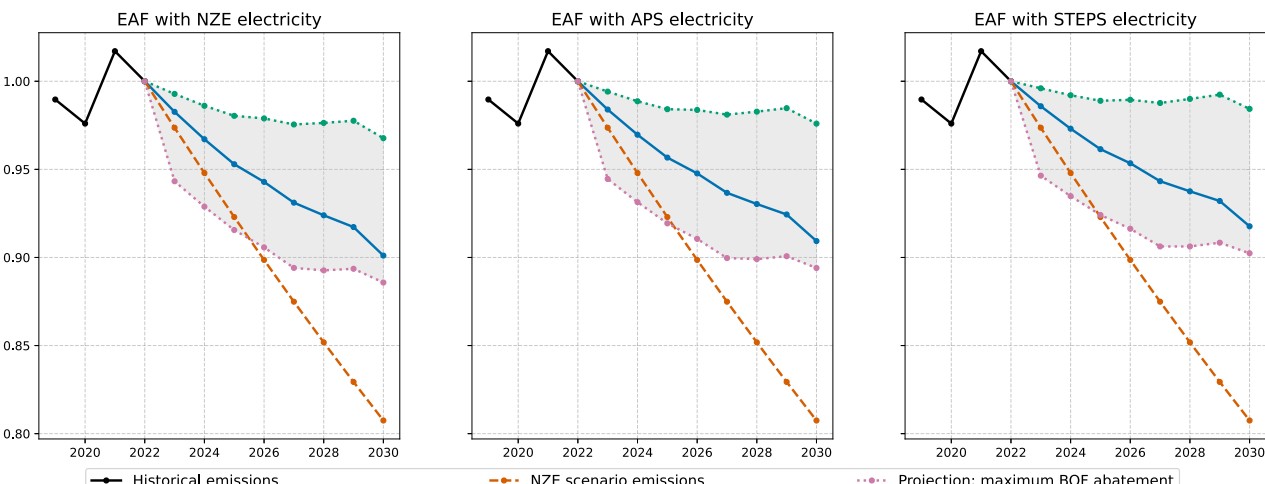

Fig. 2 | **Projected bottom-up emissions of the Iron & steel sector under different assumptions on the future evolution of emission factors, normalized to 1 in 2022.** In the left chart, the EAF (Electric Arc Furnace) emission factor is computed assuming that the carbon intensity of electricity production follows the NZE (Net Zero) scenario, in the middle chart the APS (Announced Pledges) scenario, and in the right chart the STEPS (States Policies) scenario. The upper dotted curve is computed assuming that the emission factor of BF-BOF (Blast Furnace-Basic Oxygen Furnace) technology is constant, the middle solid curve uses historical extrapolation from the past 20 years, and the lower dotted curve assumes full implementation of best available technologies for the decarbonization of the BF-BOF process (see "Methods" section for details). Even in the most optimistic case in which the electricity sector achieves the net zero target, and all available abatement technologies are fully implemented (left chart, lower dotted curve), we find a substantial deviation (10%) in the emissions of the steel sector compared to its own net zero target. In the least optimistic case, where the BF-BOF emission factor stays constant, and the electricity carbon intensity follows the STEPS scenario rate, the deviation increases to 22%.

which decarbonization targets are available. We then aggregate the published targets for these companies and compare the target decarbonization rates with the actual decarbonization rates from asset-level data.

### Evaluating the net-zero alignment of the iron and steel sector
We find that the iron and steel sector as a whole is not aligned with the IEA Net Zero 2050 scenario. Figure 2 compares the bottom-up (asset-level) emission trajectories for the entire iron and steel sector with the target emission trajectories for the net zero scenario. To ensure coherence between the global steel production trajectory and the reference emission trajectory, we assume that steel production also follows the IEA Net Zero 2050 scenario trajectory.

The three graphs and the projected emission curves within each graph differ in the assumptions made to determine the future emission factors of the main steel production routes, Electric Arc Furnace (hereafter EAF) and Blast Furnace-Basic Oxygen Furnace (hereafter BF-BOF or simply BOF). Namely, in the left graph, the carbon intensity of electricity for EAF emission factors is taken from the NZE scenario, in the right graph, it is taken from the APS scenario and in the bottom graph from the STEPS scenario. Note that the three IEA scenarios are not directly comparable with each other: while the APS and STEPS datasets include regional disaggregation, the NZE scenario is published only at the global level.

In each graph, the red curve assumes a constant emission factor for the BF-BOF technology, the orange curve uses an emission factor extrapolated by linear regression from the past 20 years, while the blue curve represents the emission factor under full implementation of best available technologies for decarbonizing the BF-BOF process (see "Methods" section for details).

Even in the most optimistic case, where the electricity sector achieves the net zero target, and all available abatement technologies are fully implemented (left chart, orange curve), the iron and steel sector overshoots its net zero target by 10%. In the least optimistic

case, where the BF-BOF emission factor stays constant, and the electricity carbon intensity follows the STEPS scenario rate, the overshoot increases to 22%.

### Comparing bottom-up emission projections with stated targets
Based on estimates from asset-level data, we find that the decarbonization pathways of companies from the iron and steel sector, which publish decarbonization targets, are, on average, not aligned with those targets. Figure 3, for the subset of companies with reported stated targets, compares the future emissions computed from stated targets with future emissions estimated from asset level data. Since in this case the reference emission trajectory is given by the company's stated target rather than taken from a scenario, we ensure coherence between future carbon intensity of electricity intensity and future global steel production, by taking them from the same scenario. In the left graph, the future global steel production and the carbon intensity of electricity production are given by the NZE scenario, in the middle graph by the APS scenario, and in the right graph by the STEPS scenario. Similarly to Fig. 2, in each graph, the red curve assumes a constant emission factor for the BF-BOF technology, the orange curve uses an emission factor extrapolated from the past 20 years, while the blue curve represents the emission factor under full implementation of best available technologies.

We find that even under the most optimistic assumption about the future evolution of the BF-BOF emission factor, the companies substantially overshoot their stated targets with respective differences of 15%, 21% and 28% for the NZE, APS, and STEPS scenarios.

Note that the targets examined in this paper, as well as the IEA emission scenarios used for comparison, are formulated in terms of annual emissions, whereas IPCC carbon budgets are defined cumulatively. The use of annual emission targets is standard in the scenario literature, as it allows one to track the time profile of sectoral emission pathways and to analyze the dynamics of transition in the iron and steel sector.

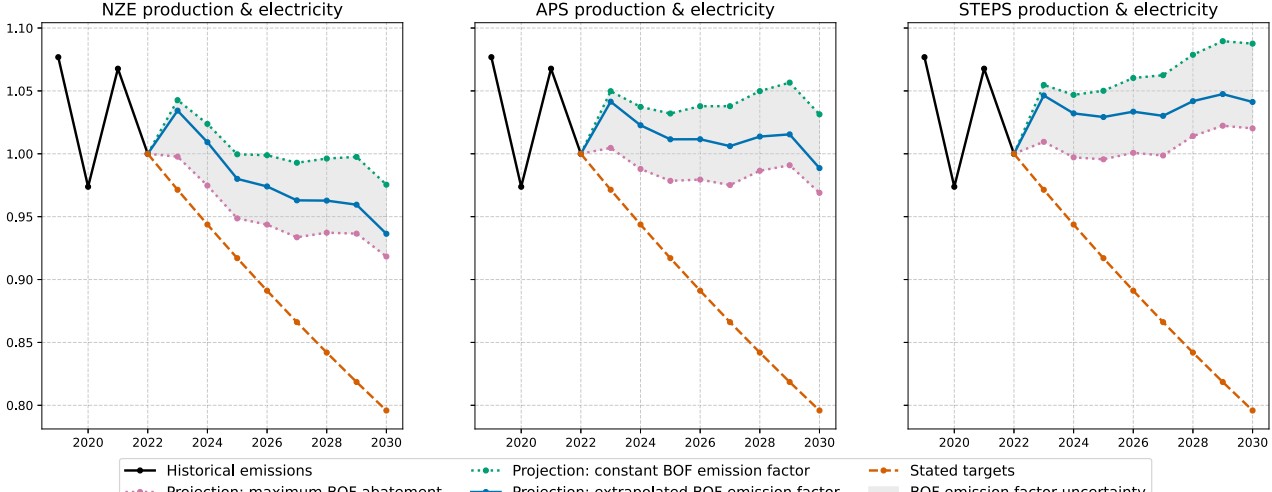

**Fig. 3 | Comparison of stated emissions with bottom-up projections on a subsample of 36 companies for which targets in 2030 were available.** This subsample represents ~24% of global capacity in 2022. In the left graph, projections are computed assuming that the carbon intensity of electricity production and the global steel production follow the IEA STEPS (Stated Policies) scenario trajectory; in the middle graph the they follow the IEA APS (Announced Pledges) scenario, and in the right graph, they follow the IEA NZE (Net Zero) scenario. The upper dotted line is computed assuming that the emission factor of BF-BOF (Blast Furnace-Basic Oxygen Furnace) technology is constant, the middle solid curve uses historical extrapolation from the past 20 years, and the lower dotted curve assumes full implementation of best available technologies for the decarbonization of the BF-BOF process.

## Discussion

Our bottom-up, scenario-based methodology provides complementary and independent estimates of future companies' emissions trajectories. This science-based and transparent way to assess companies' progress on emissions reduction vis-à-vis companies' stated targets is important to understand if, and which, firms and sectors are contributing to the net zero scenarios, and if they are delivering on their alignment promises. This reality check, in turn, is important for supervisory bodies to improve the relevance of disclosure standards worldwide and the effectiveness of controls. Indeed, while an increasing number of companies have set emissions reduction targets (as of April 2024, there are 5121 companies with SBTI-validated targets), most companies remain misaligned with Paris agreement pathways[8,18]. Besides ambiguity in targets' definitions[19,20], an important reason for such misalignment may be the backward-looking nature of the alignment assessment methodologies. These are mainly based on past carbon performance of companies and fail to take into account future emission reductions, which are already visible in companies' assets. By developing a forward-looking methodology for emission projection based on asset-level data, our paper contributes to reducing the gap between the Paris agreement goals and the actual emission pathways.

Our paper contributes to the literature on committed emissions, which has, so far, mainly addressed emissions from fossil fuels[21] and energy infrastructure[22–24]. While in this work we have focused on the steel sector, our methodology could be applied to other sectors as well, in particular those for which production is carried out by means of a set of alternatives technologies for which emission intensity is known relatively well, such as cement, aluminum, pulp & paper, or electricity generation (see Supplementary Section 3 and Supplementary Table 3 for details on specific challenges and data sources for each sector). Further, it could also be applied to other sectors, such as motor vehicle manufacturing, for which most emissions are indirect (e.g., Scope 3 according to the GHG Protocol) but can be estimated relatively reliably based, e.g., on emission intensity factors associated with use of the product (e.g., $CO_2$e grams per kilometer).

Finally, our results are important for the construction of net-zero investment portfolios. In climate finance, the notion of "financed emissions" refers to cumulated shares of emissions associated with the economic activities financed by a portfolio. However, emissions used in the calculations are the historical ones. Moving to a forward-looking metric of emissions is crucial to direct investment flows to activities that contribute to the greening of the economy[25].

The following sources of uncertainty affect the estimates provided by our methodology. First, the scenarios used for future steel production and electricity carbon intensity have a large impact on the estimates. For this reason, we provide results conditional on the scenario selection. Second, companies can have multiple production facilities with different technologies and emission intensities. Thus, the future utilization rates of the various technologies are important. We have tested the robustness of our results to a range of assumptions in this regard. Third, the future carbon intensity of specific steel production technologies could decrease over time. To assess the impact of this source of uncertainty on our estimates, we have developed a hybrid approach, combining statistical projections of past trends for emission factor evolution with estimates of the maximum abatement potential of best available technologies and data on their current deployment at the country level. Fourth, commissioning/decommissioning dates of plants affect the calculation of companies' emissions. However, large companies tend to have several dozen facilities, so we expect random errors on these dates to compensate for each other.

A more detailed discussion of the sources of uncertainty is reported in Supplementary Section 2. The above sources of uncertainty limit, at this stage, the applicability of the methodology to the short and medium term (i.e., until 2030). Nevertheless, the medium-term is the crucial horizon for the feasibility of decarbonization, and our methodology contributes to assessing individual companies' actions in this regard.

The main limitations of this study stem from the use of country-level emission factors for steel production and the limited availability of data on the current and future deployment of best

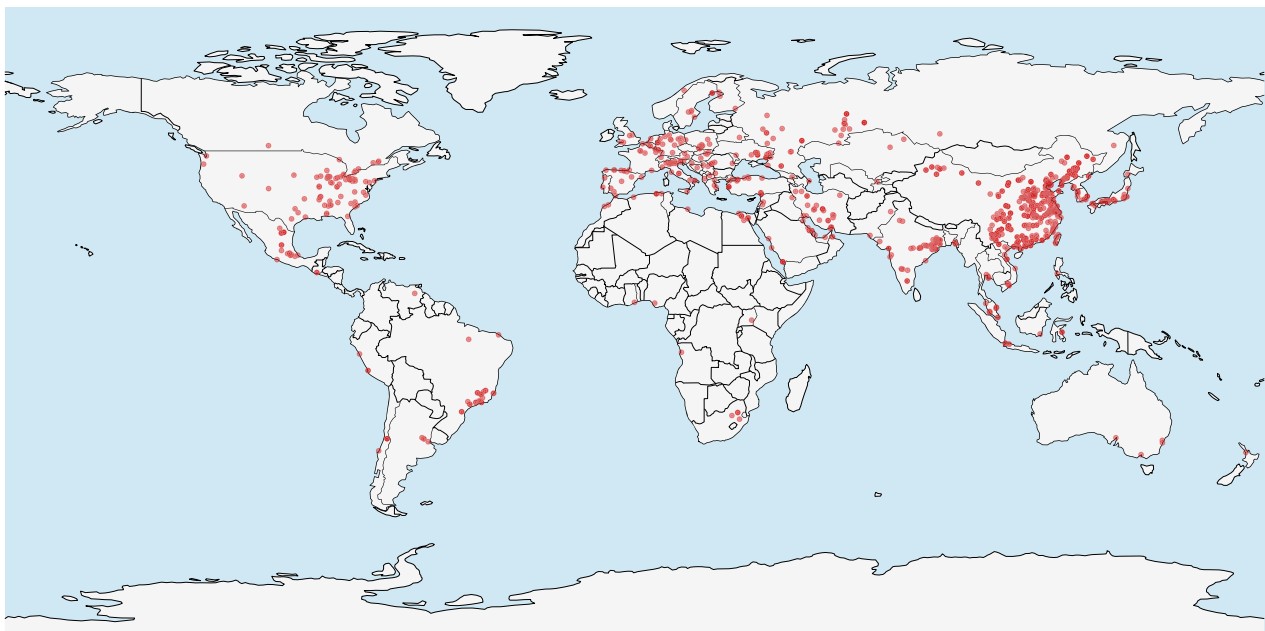

**Fig. 4 | Locations of operating plants from the Global Iron and Steel Tracker database in 2021.** Capacity is mainly concentrated in the following regions: Asia–Pacific (China having ~50% of global capacity), Europe and North America.

available technologies. Enhancing the coverage, consistency, and transparency of open asset-level datasets in the steel industry-and in other emission-intensive sectors-would substantially improve the accuracy and robustness of future bottom-up alignment analyses.

We acknowledge that the current geopolitical context increases uncertainty both around scenarios of decarbonization of the electricity sector, as well as on the cost of best-available technologies for BF-BOF steel production. The goal of the paper is to provide a method to assess the potential discrepancy between current firms' stated emissions and independent estimates of emissions under reference scenarios developed by international agencies (i.e., IEA). The elaboration of alternative scenarios that would take into account geopolitical factors is beyond the scope of this work.

Overall, our results matter for the design of more targeted climate and industrial policies, which would lead to a more effective use of public resources, especially in times of tight budget constraints.

## Methods
### Data sources
Our framework relies on four types of data sources (see Fig. 1): global and sector-specific scenario data; asset-level data; technology data; reported data on GHG emissions and company-level decarbonization targets.

### Scenario data
For global scenario data, we use the World Energy Outlook 2023 scenarios from the IEA (see iea.org/reports/world-energy-outlook-2023), including NZE, APS and STEPS. They include values for global production, emissions, electricity intensity and carbon prices. The IEA provides country-level scenario data only for STEPS and APS. For the NZE scenario, we use global-level values.

For sector-specific scenario data, we use the Energy and Technology Perspectives report on Iron & Steel (see iea.org/reports/energy-technology-perspectives-2023) from the IEA, in order to derive sector-specific values (such as the historical level of emissions, which serves as a starting point for describing emissions trajectories).

### Asset-level data
The Global Iron and Steel Tracker (previously known as Global Steel Plant Tracker, see globalenergymonitor.org/projects/global-iron-and-steel-tracker/) database tracks over 950 plants owned by over 600 companies and representing an annual crude steel capacity of 2.9 billion tons (94% of the total). For each plant, it includes plant capacity, production, development status (operating, construction, ...), main production method (BF-BOF, EAF) and ownership fraction. The start year for projected plants is also available, allowing for asset-level emission projection. Figure 4 plots the location of plants on a map.

Opening dates of future plants are well represented in the database; however, closure dates of some plants may not be fully represented. For this reason, we add closure dates for the oldest blast furnace plants to the database, separately for each scenario to meet a global scenario-dependent capacity constraint, coherently with the following approach:

- Global capacity is projected based on the latest historical value and the annualized growth rate of global production from a given scenario.
- At each future date, we compare bottom-up global capacity with the above projection, and shut down the oldest blast-furnace plants whenever the constraint is not satisfied. We apply this procedure at the global level for all plants, disregarding specific geographic or firm-level capacity constraints.

### Technology data: GHG emission factors
The main technologies for crude steel production are Blast Furnace (for converting iron ore into pig iron) combined with Basic Oxygen Furnace (for converting pig iron into steel), abbreviated as BF-BOF, and Electric Arc Furnace (EAF). We use country-level emission factors from ref. 26 (see ourenergypolicy.org/resources/steel-climate-impact-an-international-benchmarking-of-energy-and-co2-intensities/), calculated with 2019 data. This study covers 16 major steel-producing countries/regions, accounting for 87% of global steel production and 91% of global steel production through the BF-BOF route, which is responsible for most carbon emissions. For the remaining countries, we use global emission factors from the World Steel Association (see

**Table 1 | Global emission factors table for several steel production processes included in the Global Iron and Steel Tracker database**

| Technology | Emission factor |
|---|---|
| electric (EAF) | 0.67 ($t$ $CO_2$/$t$ steel) |
| integrated (BF-BOF) | 2.32 ($t$ $CO_2$/$t$ steel) |
| integrated (DRI) | 1.65 ($t$ $CO_2$/$t$ steel) |
| mixed or other | 2.32 ($t$ $CO_2$/$t$ steel) |

*EAF* Electric Arc Furnace, *BF-BOF* Blast Furnace-Basic Oxygen Furnace, *DRI* Direct Reduced Iron.

**Table 2 | Best available technologies for BF-BOF (Blast Furnace-Basic Oxygen Furnace) steel production route and their $CO_2$ abatement potential, based on ref. 28**

| Process | Technology | $CO_2$ abatement (kg$CO_2$/tonne) |
|---|---|---|
| Coking | Coke dry quenching (CDQ) | 50.05 |
| | Coal moisture control (CMC) | 16.32 |
| Sinter | Heat recovery from sintering and sinter cooler | 73.83 |
| Blast furnace | Pulverized coal injection | 60.41 |
| | Top-pressure recovery turbines (TRT) | 27.3 |
| | Recovery of BF gas | 5.84 |
| | Preheating of fuel and air for the hot blast stove | 26.03 |
| BOF | Recovery of BOF gas and sensible heat | 48.19 |
| | Flue gas waste heat recovery | 19.24 |
| Casting | Continuous casting | 36.35 |
| Hot rolling | Recuperative or regenerative burner | 46.5 |
| | Process control in hot strip mill | 25.61 |
| | Waste heat recovery from cooling water | 2.95 |
| Cold rolling | Heat recovery on the annealing line | 30.06 |
| | Automated monitoring and targeting systems | 24.04 |

worldsteel.org/steel-topics/sustainability/sustainability-indicators/), calculated from 2021 data and reported in Table 1. It should be noted that steel production emission factors are highly uncertain as they depend on the age and other features of the specific plant, electricity generation energy mix, etc. For plants with mixed or unknown technology, we use the highest emission factor available for the sector at the country or global level.

Future emission factors of various steel-making technologies are likely to evolve due to decarbonization efforts of companies. For the BF-BOF route, future emission reductions depend strongly on the progress and adoption of energy-efficiency and decarbonization technologies. Historical evidence shows that carbon emissions from BF-BOF steel production have declined steadily in recent decades owing to continuous process improvements[27], and this downward trend is likely to continue. In the short term (up to 2030), emission reductions are likely to come from upgrades to Best Available Technology (BAT) and improved process control rather than from emerging new technologies with low TRL. Table 2 below lists the principal BAT options for the BF-BOF steel production route together with their estimated abatement potentials, following ref. 28 (see also ref. 29 for an earlier study).

The cumulative abatement potential of all technologies amounts to approximately 493 kg $CO_2$ per tonne of crude steel. However, plant-specific and country-specific information on the current deployment of these technologies—and on future deployment plans—is often incomplete. We therefore adopt a hybrid approach that combines technology-based and statistical estimates.

First, we estimate near-term emission-reduction trends using historical country-level data for scope-1 emissions from steel production over 2000–2019, provided to us by the authors of ref. 27. For each country, we compute the trend in the BF-BOF emission factor and its statistical confidence bounds. The results (Supplementary Fig. 3) show consistently declining emission factors across most producing countries, with faster reductions in developing regions and slower ones in developed economies, as expected.

We next estimate the technology-based emission reduction potential for each country in our database. Where available, we use data on the 2020 deployment levels of the BAT technologies listed in Table 2. Supplementary Table 4 lists the deployment rates of these technologies for major steel-producing countries. If no information is available for a specific technology or country, the deployment level is set to zero. The maximum remaining abatement potential for each technology is then computed as

remaining potential = (100% − 2020 deployment) × max potential,

where the maximum potential corresponds to the values in Table 2. Summing across technologies yields the remaining abatement potential for each country.

We finally update the statistical confidence bounds of the first step, assuming that the emission factor cannot decrease below the maximum cumulative abatement potential achievable by full BAT

deployment over the same period. Accordingly, if the statistical lower bound falls below the value implied by complete BAT deployment, it is set to that lower limit.

The principal figures in the paper include uncertainty ranges based on these updated confidence bounds, which incorporate both observed historical trends and the quantified technological abatement potentials. More specifically, in each graph of Figs. 2 and 3, the red curve assumes a constant emission factor for the BF-BOF technology, the orange curve uses an emission factor extrapolated using the trend estimated on 2000–2019 data, while the blue curve represents the hybrid lower bound computed by combining the statistical lower bound and the lower bound assuming maximum BAT deployment as described above.

For the EAF production route, we assume that the emission factors follow the global decarbonization trend of electricity production, which is available in IEA scenarios, at the global level for the NZE scenario and at the regional level for the APS and STEPS scenarios.

### Reported emissions data and stated targets

We use reported scope 1 and (location-based) scope 2 emissions from the CDP questionnaires, and the reported scope 1 and scope 2 emissions from Refinitiv Eikon. For companies that are present in both data sources, we adopt a conservative approach by defining the top-down (reported) emissions as the maximum value between CDP and Refinitiv scope 1 and 2 emissions, for each company and year. Reported emissions are used to compare them to bottom-up emissions, at the company level, and train the statistical model used for estimating adjusted bottom-up emissions.

To this end, based on the corporate tree structure data from Refinitiv and the publicly available information, we reconstruct corporate ownership and attribute bottom-up emissions from all identified subsidiaries to the ultimate parent. We match companies from top-down and bottom-up datasets based on PermID or fuzzy matching algorithms.

We compute standardized company-level target emission reduction rates using stated emission-reduction targets from Refinitiv, the Green Steel Tracker, the Net Zero Tracker, and the Transition Pathway Initiative. Stated targets are available for 36 companies in our database, representing approximately 24% of global steelmaking capacity in 2022.

Target-setting practices vary considerably across firms, leading to potential inconsistencies in the data. Companies differ in their choice of base years, the use of absolute versus intensity-based metrics, and the extent of disclosure. Of the 36 companies we analyze, five have had their targets validated by the Science Based Targets initiative (SBTi), providing a degree of methodological assurance and comparability.

### Asset-level emission projection and alignment assessment

We develop a two-step methodology for estimating future company-level $CO_2$ emissions from asset-level data (see Fig. 1). In the first step, we estimate the raw bottom-up emissions, as follows

$$\widetilde{E}_{f,t}^{BU} = \sum_p \Pi_t^p * EF_t^p \tag{1}$$

$$= \sum_p C_t^p * UR_t^p * EF_t^p \tag{2}$$

where $\widetilde{E}_{f,t}^{BU}$ are raw bottom-up emissions of firm $f$ for a period $t$, $\Pi_t^p$ is the production value of the plant $p$ for the period $t$, $C_p^c$ is the capacity of the plant $p$ for the period $t$, $UR_p^c$ is the utilization rate for the plant $p$ for the period $t$ and $EF_t^p$ is the emission factor for the plant $p$ and period $t$. The sum is computed over all plants owned by firm $f$ in the period $t$.

For estimating historical bottom-up emissions, we use formula (1), with production values computed as explained below. For estimating future bottom-up emissions, formula (2) is used instead. This method requires information regarding:

- The list of assets of each company with their capacities and production technologies for the period $t$. This is computed from the GIST database, which provides closure dates for existing plants and opening dates for planned production sites.
- The emission factors for each technology. The emission factors are projected separately for the BF-BOF and for the electric technology, as explained above.
- The utilization rates for each plant. The projected future utilization rate is computed to match the global production from a given reference scenario according to one of the two alternative plausible assumptions: (i) country-level utilization rate; and (ii) constant market share. These two approaches are explained in detail below.

In the second step, to account for emission sources that are not directly related to the production process, such as electricity consumption for other uses than electric arc furnace steel production, we use a statistical model to correct the raw bottom-up emissions, see Equation (3).

### Historical asset-level production estimation

The Global Iron and Steel Tracker provides excellent coverage for capacity over 95 % globally and for most major producing countries. In contrast, production coverage is substantially lower. Only about 54 % of capacity has production data for the United States; the number is even lower for China, and below 75 % for most other producers (See Supplementary Figs. 4 and 5).

Because of this limitation, estimating plant-level capacity utilization directly from the GIST dataset could introduce substantial uncertainty and bias. To ensure consistency and representativeness, we therefore rely on country and technology-level utilization rates derived from World Steel Association (WSA) production data combined with GIST capacity data throughout the analysis. Figure 5

compares the country-level utilization rates for the BF-BOF steel production route computed with WSA production data with those estimated at the plant level with GIST production data. This graph shows an important bias in plant-level utilization rates of developing countries, potentially due to the better production data availability for more modern plants, which are also more likely to operate at higher utilization rates.

We thus replace the actual utilization rate of the plant $UR^p$ with the average value for the plant's category $\widetilde{UR}^p$, and compute the estimated production $\widetilde{\Pi}^p = C^p \widetilde{UR}^p$, where $C^p$ is the plant's capacity. Since bottom-up global production does not exactly match the true global production (e.g., because the GIST database does not contain all plants), in the last step, we adjust the estimated bottom-up production values by multiplying them by the ratio of true global production for the given year to estimated bottom-up production.

### Projecting asset-level production

We use two alternative methods to project future utilization rates and production values at the plant level, based on global production from IEA scenarios. Both methods lead to very similar results.

- Method 1: country-level utilization rate: We calculate historical country-level utilization rates based on 2022 production data and normalize them with a common global factor to match the global steel production of the given scenario $\Pi_t^s$:

$$\widetilde{UR}_t^c = \widetilde{UR}_0^c \frac{\Pi_t^s}{\sum_c C_t^c \widetilde{UR}_0^c},$$

  where $C_t^c$ denotes the total steel-making capacity of country $c$ at time $t$. For this method, there is a small number of plants for which the historical country-level utilization rate is zero due to low data quality. We assume all plants will be operating in the future, so we impute the median global utilization rate for these plants.

- Method 2: constant market share: For each firm $f$, we first compute the market share, based on 2022 production data, which either comes from the World Steel Association Reports, when such information is available, or is estimated from the Global Iron and Steel Tracker Database. Then, assuming that the market share stays constant, we estimate future production $\Pi_t^f$ from global production $\Pi_t^s$ in the given reference scenario. Finally, we assume each plant of this firm to have the same utilization rate defined as:

$$\widetilde{UR}_t^f = \frac{\Pi_t^f}{C_t^f} = \frac{\Pi_t^s}{C_t^f} \frac{\Pi_0^f}{\Pi_0},$$

  where $C_t^f$ denotes the capacity of the firm $f$.

- Upper and lower bounds on emissions: In addition to the two reference methods, we compute the upper and lower bounds on future emissions. To this end, we first determine the future production of each country based on global projected production from scenarios, assuming a constant market share for each country (based on 2022 values). Then, for the upper bound, we assume that in each country, the steel is produced by the most polluting plants (with the highest carbon intensity), and for the lower bound, we consider that the most carbon-efficient plants are used to match the scenario-based production estimate. In the rare occurrence of future plants opening in countries with no pre-existing market share, we assume all plants operate at full capacity.

Figure 6 shows the impact of utilization rate uncertainty on the projected emissions of the iron and steel sector. This graph can be directly compared to the left graph in Fig. 2: the orange curve (central

**Fig. 5 | Comparison of country-level BF-BOF (Blast Furnace-Basic Oxygen Furnace) utilization rates computed with World Steel Association data (dashed lines) with plant-level BF-BOF utilization rates computed with Global Iron and Steel Tracker (GIST) data (vertical bars).** Error bars correspond to one standard deviation of plant-level rates. Numbers on the bars correspond to the number of plants for which production data is available. For China, the GIST-based utilization rate in 2019 and 2020 is slightly higher than one because the actual production of some plants is higher than their declared capacity due to temporary capacity overload, which is common in China. This graph shows an important bias in plant-level utilization rates due to the better production data availability for more modern plants, which are also likely to operate at higher utilization rates.

estimate) is the same in both graphs; the pink shaded area in Fig. 2 shows the uncertainty due to possible evolution of the BF-BOF emission factor, while the blue shaded area in Fig. 6 highlights the uncertainty due to possible evolution of plant utilization rates. We see that the utilization rate uncertainty can modify the 2030 emission estimate by at most 5% which does not qualitatively change the conclusions of our study.

**Estimating adjusted bottom-up emissions**

Out of over 600 companies represented in the GIST database, there are 50 that report their emissions, representing around 25% of the global steelmaking capacity. This number is relatively low because many smaller steel companies, especially those located in countries with low reporting standards, do not report their emissions. The self-reported emissions from these 50 companies can be used to (1) validate the methodology and (2) correct some biases in bottom-up emissions. Figure 7 plots the self-reported firm-level emissions $E_f^{TD}$ (also referred to as top-down emissions) for these companies for the year 2021 as a function of the raw bottom-up emissions $\widetilde{E}_f^{BU}$. This graph shows that (1) the points are close to the diagonal and (2) top-down emissions may nevertheless deviate from bottom-up estimates due to uncertainty of emissions factors and other emission sources, not directly explained by the basic production process, which are reflected in self-reported emissions but not in bottom-up estimates.

To improve our emission estimates, we build a statistical model for the reported company-level emissions $E_f^{TD}$, which uses the raw bottom-up emissions $\widetilde{E}_f^{BU}$ as an explanatory variable. The model is specified as follows:

$$\log E_{f,t}^{TD} = \beta_0 + \beta_1 \log \widetilde{E}_{f,t}^{BU} + \epsilon_{f,t}, \qquad (3)$$

where $E_{f,y}^{TD}$ denotes the top-down emissions for firm $f$ and period (year) $t$, $\widetilde{E}_{f,t}^{BU}$ denotes the corresponding raw bottom-up estimate, $\beta_0$ and $\beta_1$ are coefficients and $\varepsilon_{f,t}$ is the noise term. The model is fitted on historical data spanning the 2019–2021 period, for which both bottom-up and top-down data are available after removing outliers as described in the following paragraph. Fitting the model (3) on the remaining ones, we obtain estimated intercept value $\widehat{\beta}_0 = -0.0826$, estimated slope value $\widehat{\beta}_1 = 0.9964$ and coefficient of determination $R^2 = 0.964$. On the one hand, these results, in particular, slope and $R^2$ values, which are both quite close to one, validate our bottom-up estimates. On the other hand, we consider that after removing outliers, the self-reported historical emissions faithfully reflect the actual carbon emissions of each company. Assuming that the relationship between bottom-up and top-down emissions remains stable in time, this linear model can therefore be used to improve our estimates of future emissions. We therefore define adjusted bottom-up emissions

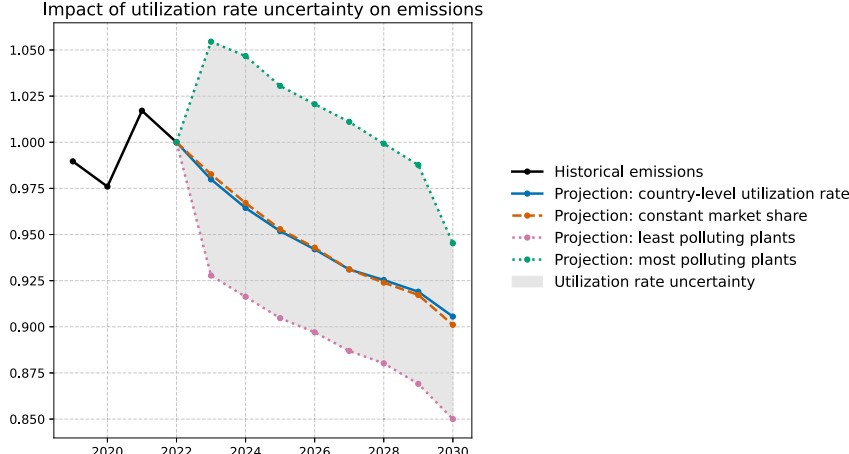

**Fig. 6 | Projected steel sector emissions under different assumptions on the future utilization rates of steel plants.** This graph corresponds to the net-zero scenario for carbon intensity of electricity and for global steel production. The BF-BOF emission factor is obtained by historical extrapolation from the past 20 years. The dashed curve computes the utilization rate of each plant using the constant market share method, the solid curve uses the country-level utilization rate method, the lower dotted curve uses the least polluting plants in each country to match the projected country-level production, and the upper dotted curve uses the most polluting plants.

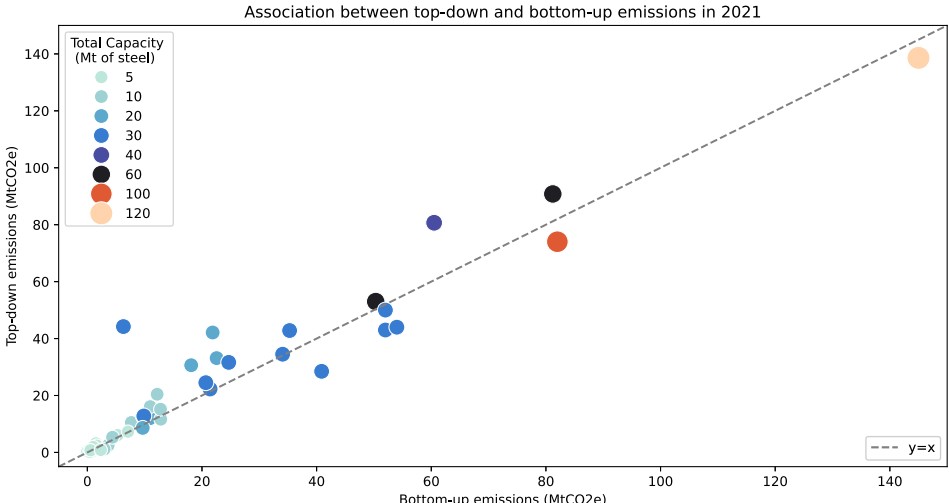

**Fig. 7 | Association between top-down and raw bottom-up emissions.** The plot shows top-down (reported) and bottom-up (estimated using our methodology) emissions for a selection of steel plants. Circles correspond to individual steel plants, with size and color representing their capacity, as shown in the legend. The dashed line indicates the situation when the two methods give the same result.

for firm $f$ and future period $t$ using

$$E_{f,t}^{BU} = \widehat{\beta}_0 + \widehat{\beta}_1 \widetilde{E}_{f,t}^{BU},$$

where $\widetilde{E}_{f,t}^{BU}$ is the raw bottom-up emissions estimate given by formula (2) and $\widehat{\beta}_0$ and $\widehat{\beta}_1$ are estimated values of, respectively, intercept and slope of regression (3).

### Treatment of outliers

Since for some companies, reported emissions are not available for the entire 3-year period, 50 companies with self-reported emissions over 3 years of data yield a total of 137 data points. Some of these data points are considered to be outliers. On the one hand, when top-down emissions are substantially larger than bottom-up emissions, it may indicate that a company's main operation is not steelmaking. On the other hand, when bottom-up emissions are substantially larger than top-down emissions, it may indicate misreported top-down emissions

(especially for smaller companies), or a company mismatch (e.g., a firm actually belongs to a higher-level company and its emissions should be attributed to the latter one). We remove outliers using the IQR method[30], that is, we consider outliers to be all data points for which the inequalities

$$Q_1 - \frac{3}{2}\text{IQR} < \log(E_{f,t}^{TD}/\widetilde{E}_{f,t}^{BU}) < Q_3 + \frac{3}{2}\text{IQR}$$

are not satisfied, where $Q_1$ and $Q_3$ are first and third quartiles of data and $\text{IQR} := Q_3 - Q_1$ is the interquantile range. This method removes 18 data points out of 137.

### Estimating stated emissions trajectories

For companies with reported reduction rates related to the given baseline and target years, we derive the stated emissions in 2030 as follows:

- We calculate 2030 emissions based on announced reduction rates and top-down (company-reported) emissions from Refinitiv, and from company sustainability reports for the reported baseline year. When top-down emissions are unavailable for that baseline year, we use values from the closest year (e.g., 2018 emissions for a 2017 reported baseline). When reported baseline years are unavailable, we replace them with the year 2019, being a pre-COVID year with a good coverage of top-down emissions.
- We calculate the annualized emissions growth rate between the baseline year and 2030 for each company.
- We then apply annualized growth rates to 2022 bottom-up emissions onwards, in order to deduce the stated target trajectory, and to make historical and projected trajectories comparable.

## Data availability

*Input data*: Forward-looking data (growth rates of steel production, decarbonization slopes of electricity generation) are taken from the World Energy Outlook 2023 dataset (iea.org/data-and-statistics/data-product/world-energy-outlook-2023-free-dataset-2). The free version provides aggregated values, and the extended version provides more granular values at the regional level. Our results are based on the latter. Plant-specific data (capacity, technology, opening and closing dates…) used in this study are available from Global Iron and Steel Tracker (globalenergymonitor.org/projects/global-iron-and-steel-tracker/) free of charge. We use country-level emission factors from ref. 31 and global emission factors from the World Steel Association 2023 Sustainability Indicators report (worldsteel.org/steel-topics/sustainability/sustainability-indicators-2023-report/). For estimating plant-level production, we use country-specific utilization rates, derived for each country from production data from the World Steel in Figures report of the World Steel Association (worldsteel.org/publications/bookshop/world-steel-in-figures-2023/) and capacity data from the OECD. Lastly, we use reported emissions from both the Carbon Disclosure Project (cdp.net/en/investor/ghg-emissions-dataset) and Refinitiv as top-down emissions for calibrating our model. We use corporate stated targets from Refinitiv, the Net Zero Steel Tracker (zerotracker.net/) and the Green Steel Tracker (industrytransition.org/green-steel-tracker/). *Output data*: the data generated in this study (historical and projected company-level emissions) are available free of charge and without access restrictions from the Pladifes database (pladifes.institutlouisbachelier.org/bottom-up/).

## Code availability

The code to reproduce our results and plot the main graphs is publicly available from Zenodo[32] with the permanent https://doi.org/10.5281/zenodo.18508837.

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

## Acknowledgements

The work of H.S., T.B., and P.T., and the purchases of the data sets for this project were financed by Agence Nationale de Recherche via the Pladifes project (ANR-21-ESRE-0036). IM acknowledges the financial contribution of (i) the European Commission's TSI 2023 Flagship Technical Support Project 'ESG risk management framework for the financial sector (ESG UPTAKE)' and (ii) the European Commission's Horizon Europe project Nature-3B: including Nature in decision making of central Banks, investment Benchmarks & Bond issuers, Grant agreement ID: 101182455.

## Author contributions

H.S.: methodology, software, data curation, writing—original draft, S.B.: conceptualization, methodology, writing: review and editing, I.M.: conceptualization, methodology, writing: review and editing, T.B.: methodology, validation, P.T.: conceptualization, methodology, writing: review and editing, supervision.

## Competing interests

The authors declare no competing interests.
