## [Transparent Peer Review File · Nature Communications]

Estimating firms' emissions from asset level data helps revealing (mis)alignment to net zero targets

Corresponding Author: Professor Peter Tankov

Version 0:

Reviewer comments:

Reviewer #1

(Remarks to the Author)

This paper presents a bottom-up estimation of future emissions in the steel industry, utilizing factory-level operational forecasts derived from asset conditions. This methodology, which is based on estimates derived from broad classifications of plants, their operational status, and operational feasibility (closure etc.), is primarily employed at the company, limited regional, or national level. Consequently, it may be perceived as lacking novelty for academic publications in this regard. However, the strength of this work lies in its innovative expansion of this bottom-up approach to a global scale. To elucidate future emissions in a specific sector (steel) on a global level, the selection and processing of data, integration of diverse data sources, and addressing challenges unique to a broad scope are commendable aspects of this study.

To further enhance the value of this paper, which strengths lie in the expansion of evaluation to a global scale, I encourage the authors to connect this global assessment to a fair aggregation of each company's performance within the global context. Addressing the following concerns will help strengthen this connection.

The primary concern pertains to the apparent contradiction between the assertion that the methodology can predict future emissions and the insufficient consideration of future technological advancements.

The emissions are derived from production volume and emission factors; however, the setting of these crucial emission factors appears imprecise. The calculations that utilize varying values for each company's plant primarily account for on-off operations, operating rates, and major technological types, such as between blast furnaces (BF) and electric furnaces, while assuming constant emission factors for each technology. However, the perspectives regarding energy-saving and decarbonization technologies, as well as the levels of technological advancement and future developments for each country and company, are not sufficiently addressed. In reality, these factors reflect the differences in corporate efforts and regional characteristics (both natural and social conditions). If technological advancements are not incorporated, there is a risk of overestimating emissions for the entire steel sector, which may weaken the paper's argument regarding discrepancies with the IEA Net Zero 2050 scenario. Furthermore, the title "Estimating firms' emissions from asset level data" may be misleading if it fails to account for the differences in technological advancements among companies as previously mentioned.

Below is a summary of opinions, acknowledging that some points may overlap with the aforementioned concerns:

The paper states that national-level emission factors are employed, with the World Steel Association's factors utilized in their absence. Although this study endeavors to enhance prediction accuracy by leveraging the Global Steel Plant Tracker database to gather detailed information on over 950 plants, there appears to be a lack of alignment between these emission factors and actual conditions, which warrants to be improved. The progress in emission reduction varies significantly based on the advancement and adoption of energy efficiency and decarbonization technologies, leading to differing emission factors. It is crucial to systematically organize the factors that significantly impact emission factors and to quantitatively demonstrate how these differences influence the results. Particularly for future emission factors of technologies such as BF-BOF, where various energy efficiency and decarbonization technologies can be applied, it is not reasonable to rely solely on historical values or fixed numbers. As previously mentioned, in addition to technological advancements, the degree of technology adoption and diffusion also influences emission factors. If stick to the methodology of historical trends, both in technological progress and in technology adoption and diffusion need to be thoroughly examined, and each requires careful scrutiny. Given that detailed data for each company is utilized, incorporating these aspects could enhance the novelty of the study. Furthermore, there are increasing opportunities for energy-saving technologies and decarbonisation initiatives to be disclosed in corporate reports, such as sustainability reports and websites with their investment plans, and it is believed that

recent advancements in AI and text mining can facilitate the verification of this content. It would also be beneficial to mention the scope of impact related to changes in policies, raw material and fuel procurement, and recent movements in energy security, even if a comprehensive analysis cannot be conducted within this paper.

The paper employs global-level values from the NZE scenarios while also incorporating regional-level values from the APS and STEPS scenarios. However, the lack of consistency in the application of global and regional scenario data has not been discussed, and it would be prudent to address how this inconsistency impacts the results.

The statement "We always use the most granular geographical level for which data is available" is acknowledged; however, while this may be the best available option, it is a matter that requires careful consideration throughout the entire context. For instance, companies and plants lacking data are often situated in developing regions where data organization is delayed or among companies with limited financial resources. In such instances, the figures from a select few companies with available data may not accurately represent the region or country. If such figures are applied to the aforementioned types of companies, it could lead to either over- or underestimation. Therefore, evaluating the impact of this issue is essential. If these impacts are significant, it would be advantageous to incorporate the differences in capabilities among such companies into the evaluation. The same consideration applies to the discussion on utilization rates.

Regarding cross-sector applicability, the methodology utilized in the steel sector is suggested to be applicable to other sectors. While it is evident that the broad framework can be applied, the authors should demonstrate how the challenges in obtaining data for other sectors (such as cement, aluminum, pulp and paper, and power) differ from those encountered in the steel sector and how these differences may affect the results. In addition to the unique challenges of applying this methodology across various sectors, summarizing the availability of plant-level data (e.g., plant closures and utilization rates) in a table would be beneficial. This summary could include the degree of technological progress and the adoption/diffusion of various energy efficiency and decarbonization technologies, which the authors have not fully considered thus far. Furthermore, providing concrete examples and data to support the claim of applicability to other sectors could lead to significant enhancements. It would also be helpful to indicate instances where evaluation processes distinct from those used in this steel study are required for specific sectors.

It is recommended to thoroughly evaluate the impact of uncertainty. As previously noted, future technological advancements and their rates of adoption/diffusion are critical factors. If it is challenging to examine these within the scope of this paper, it is essential to demonstrate that their impact on the results has been adequately assessed. Clearly defining the range of uncertainty and evaluating the sensitivity of the results to these uncertainties is recommended.

Additionally, it is crucial to assess whether the assumptions regarding the carbon intensity of future power generation are realistic, as these assumptions significantly impact the results. A careful examination of the basis for the scenarios and their consistency with historical data is necessary, along with sensitivity analyses on representative factors to validate the scenarios. Recent advancements in information technology have raised the possibility of a significant increase in future electricity demand, particularly drawing attention to the energy procurement strategies of IT companies. While there are indications that this demand may be met through decarbonized energy sources and nuclear power, there remains a divergence of opinions regarding how the balance of supply and demand will be forecasted across various scenarios. It would be beneficial to present these perspectives and incorporate sensitivity analyses for several scenarios to provide a more comprehensive understanding of the potential outcomes.

Discrepancies between the targets published by companies and the actual emission pathways may stem from data inconsistencies or methodological issues. To address this concern, verifying the transparency of the companies' target-setting processes and reporting requirements is essential. For instance, in my experience, discrepancies have sometimes been noted between CDP data and the data published by individual companies. There may also be instances where the compilation year is outdated. Acknowledging these issues would be prudent.

Reviewer #2

(Remarks to the Author)
Please see attached.

Version 1:

Reviewer comments:

Reviewer #1

(Remarks to the Author)

I would like to express my sincere gratitude for your detailed and thoughtful responses to my review comments. I greatly appreciate the effort you have taken to address each point thoroughly. Your revisions have significantly enhanced both the clarity and robustness of the manuscript.

Additionally, I would like to offer one further comment for your consideration. On page 11, in the section discussing "the technology-based emission reduction potential," I would like to suggest referencing previous works that are closely related to the methodology employed in this study. Specifically, there has been a study utilizing technology deployment rates (see below) that initiated similar approaches to estimating technology-based emission reduction potentials. Including such references could provide valuable context and further strengthen the scientific foundation of this section.

Thank you once again for your meticulous responses and for the opportunity to review this high-quality study. I look forward to seeing the final version of your manuscript.

Reference:

IPCC Fourth Assessment Report, Mitigation, Chapter 7, Figure 7.1 refers to:

Tanaka, K., R. Matsubishi, N. Masahiro, and H. Kudo, 2006: CO2 reduction potential by energy efficient technology in energy intensive industry. [Available at: <http://eneken.ieej.or.jp/en/data/pdf/324.pdf>]

Reviewer #2

(Remarks to the Author)

I am the second reviewer.

Thank you for the response. My only recommendation and suggestion is that the authors be more upfront and clear about this key difference. The IPCC emissions budget, which is typically the ground reference for many carbon targets, is cumulative, but that the targets examined in this paper are not, and whether or not these annual targets translate to the appropriate cumulative carbon emissions targets we need to meet to remain below the 1.5/2 degree thresholds.

Overall, I commend the authors for this paper. I wish them the best.

Estimating firms' emissions from asset level data helps revealing (mis)alignment to net zero targets

Hamada Saleh ^{*} Stefano Battiston [†] Irene Monasterolo [‡]
Thibaud Barreau [§] Peter Tankov [¶]

Response to reviewers

Comments from reviewer 1

*The primary concern pertains to the apparent contradiction between the assertion that the methodology can predict future emissions and the **insufficient***

^{*}Institut Louis Bachelier

[†]University of Zurich & University of Venice

[‡]University of Utrecht, CEPR, WU Wien

[§]Institut Louis Bachelier

[¶]ENSAE, Institut Polytechnique de Paris and Institut Louis Bachelier

consideration of future technological advancements.

The emissions are derived from production volume and emission factors; however, the setting of these crucial emission factors appears imprecise. The calculations that utilize varying values for each company's plant primarily account for on-off operations, operating rates, and major technological types, such as between blast furnaces (BF) and electric furnaces, while assuming constant emission factors for each technology. However, **the perspectives regarding energy-saving and decarbonization technologies, as well as the levels of technological advancement and future developments for each country and company, are not sufficiently addressed.** In reality, these factors reflect the differences in corporate efforts and regional characteristics (both natural and social conditions). If technological advancements are not incorporated, there is a risk of overestimating emissions for the entire steel sector, which may weaken the paper's argument regarding discrepancies with the IEA Net Zero 2050 scenario. Furthermore, the title "Estimating firms' emissions from asset level data" may be misleading if it fails to account for the differences in technological advancements among companies as previously mentioned.

We thank the reviewer for raising this important issue. In response to it, we now address the impact of the future developments in energy-saving and decarbonization technologies through both statistical and descriptive technology-based analysis. Please see the answer to the 2nd comment below.

1. *The paper states that national-level emission factors are employed, with the World Steel Association's factors utilized in their absence. Although this study endeavors to enhance prediction accuracy by leveraging the Global Iron and Steel Tracker database to gather detailed information on over 950 plants, there appears to be a lack of alignment between these emission factors and actual conditions, which warrants to be improved.*

We acknowledge this important point. The issue of alignment between emission factors used in bottom-up analysis and actual conditions is explicitly addressed in our paper by adjusting the bottom up emissions through a statistical model, see paragraph "Estimating adjusted bottom-up emission" in the Methods section. In this paragraph, using company-level data from 50 companies in our database which report their carbon emissions (either through sustainability reports on their web sites or through CDP), we (a) test the validity of our bottom-up estimates by comparing the reported emissions with bottom-up estimates and (b) train a statistical model to improve emission estimates for future projections using these two data sources (reported and bottom-up emissions). We find that both the slope value and the coefficient of determination of the linear regression of log reported emissions on log bottom-up emissions is very close to one ($\hat{\beta}_1 = 0.9964$ and $R^2 = 0.964$) which validates our bottom-up estimates.

2. *The progress in emission reduction varies significantly based on the advancement and adoption of energy efficiency and decarbonization tech-*

nologies, leading to differing emission factors. **It is crucial to systematically organize the factors that significantly impact emission factors** and to quantitatively demonstrate how these differences influence the results. Particularly for future emission factors of technologies such as BF-BOF, where various energy efficiency and decarbonization technologies can be applied, it is not reasonable to rely solely on historical values or fixed numbers. As previously mentioned, in addition to technological advancements, the degree of technology adoption and diffusion also influences emission factors. If stick to the methodology of historical trends, both in technological progress and in technology adoption and diffusion need to be thoroughly examined, and each requires careful scrutiny. Given that detailed data for each company is utilized, incorporating these aspects could enhance the novelty of the study.

We appreciate the reviewer’s thoughtful and detailed comment. We agree that emission reductions in BF-BOF steelmaking depend strongly on the progress and adoption of energy-efficiency and decarbonization technologies, and that a systematic representation of these factors is essential. Indeed, historical evidence shows that carbon emissions from BF-BOF steel production have declined steadily in recent decades owing to continuous process improvements [3], and this downward trend is likely to continue. In the short term (up to 2030), emission reductions are likely to come from upgrades to Best Available Technology (BAT) and improved process control rather than from emerging new technologies with low TRL.

To address the reviewer’s request to **systematically organize emission reduction factors**, we have added Table 1 below (now included in the revised manuscript), which lists the principal BAT options for BF-BOF steel production route together with their estimated abatement potentials, following [2].

The cumulative abatement potential of all technologies amounts to approximately 493 kg CO₂ per tonne of crude steel. However, plant-specific and country-specific information on the current deployment of these technologies—and on future deployment plans—is often incomplete. We therefore adopt a **hybrid approach** that combines technology-based and statistical estimates.

First, we estimate near-term emission-reduction trends using historical country-level data for scope-1 emissions from steel production over 2000–2019, provided to us by the authors of [3]. For each country, we compute the trend in the BF-BOF emission factor and its statistical confidence bounds. The results (Figure 1, now included in the Supplementary Material) show consistently declining emission factors across most producing countries, with faster reductions in developing regions and slower ones in developed economies, as expected.

We next estimate the technology-based emission reduction potential for each country in our database. Where available, we use data on the 2020

deployment levels of the BAT technologies listed in Table 1. Table 4 in Supplementary material lists the deployment rates of these technologies for major steel producing countries. If no information is available for a specific technology or country, the deployment level is set to zero. The maximum remaining abatement potential for each technology is then computed as

$$\text{remaining potential} = (100\% - 2020 \text{ deployment}) \times \text{max potential},$$

where the maximum potential corresponds to the values in Table 1. Summing across technologies yields the remaining abatement potential for each country.

We finally update the statistical confidence bounds of the first step assuming that the emission factor cannot decrease below the maximum cumulative abatement potential achievable by full BAT deployment over the same period. Accordingly, if the statistical lower bound falls below the value implied by complete BAT deployment, it is set to that lower limit.

The principal figures in the revised paper now include uncertainty ranges based on these **updated confidence bounds**, which incorporate both observed historical trends and the quantified technological abatement potentials.

To main changes made to address this point can be found on pages 10-11 (description of the emission factor projection methodology) and in figures 2 and 3 (new uncertainty bounds).

3. *Furthermore, there are increasing opportunities for energy-saving technologies and decarbonization initiatives to be disclosed in corporate reports, such as sustainability reports and websites with their investment plans, and it is believed that recent advancements in AI and text mining can facilitate the verification of this content.*

We acknowledge that ongoing developments in AI could facilitate the extraction of information related to firms' emissions projections from annual reports. However, it remains unclear to what extent AI could address the verification of physical measures of emissions on the ground, given that there are also agency and incentive issues. In any case, it is important to have a method to estimate firms' emissions that is independent of the firms' disclosure and it is instead grounded on the data of production plants, their technology and capacity. We expect that having such a method will remain important, notwithstanding the AI developments.

To address this comment we have mentioned this point in the last paragraph of the Introduction to better clarify the relevance of our contribution.

4. *It would also be beneficial to mention the scope of impact related to changes in policies, raw material and fuel procurement, and recent movements in energy security, even if a comprehensive analysis cannot be conducted within this paper.*

We acknowledge that the current geopolitical context creates uncertainty around the future levels of decarbonization of the electricity sector, in particular because of evolving fuel procurement policies. Moreover, it increases also the uncertainty around the cost of alternative technologies for production of steel based on the BOF pathway, as listed in Table 2 of the Section Methods of the main paper, because of access to raw materials. However, our goal in the paper is to assess the potential discrepancy between current firms' stated emissions and independent estimates of emissions under reference scenarios developed by international agencies (i.e. IEA). The elaboration of alternative scenarios that would take into account geopolitical factors is beyond the scope of this work. In our view, even in the current context of uncertainty regarding climate policies, our method can inform investors' decisions to allocate their limited capital in firms with more credible decarbonization plans. We have addressed this comment by adding a sentence in the discussion section.

5. *The paper employs global-level values from the NZE scenarios while also incorporating regional-level values from the APS and STEPS scenarios. However, the **lack of consistency in the application of global and regional scenario data has not been discussed**, and it would be prudent to address how this inconsistency impacts the results.*

We appreciate the reviewer's observation. Indeed, an inconsistency exists among the three IEA scenarios: while the APS and STEPS datasets include regional disaggregation, the NZE scenario is published only at the global level. Because these scenarios are widely used and form an established reference in both industry and academic studies, we retained all three for consistency with standard practice, despite this limitation. In the revised version of the paper, we explicitly acknowledge this inconsistency and clarify that, for this reason, NZE results are not directly comparable with those from the APS and STEPS scenarios (see page 6, lines 4–7).

6. *The statement "We always use the most granular geographical level for which data is available" is acknowledged; however, while this may be the best available option, it is a matter that requires careful consideration throughout the entire context. For instance, companies and plants lacking data are often situated in developing regions where data organization is delayed or among companies with limited financial resources. In such instances, the figures from a select few companies with available data may not accurately represent the region or country. If such figures are applied to the aforementioned types of companies, it could lead to either over- or underestimation. Therefore, evaluating the impact of this issue is essential. If these impacts are significant, it would be advantageous to incorporate the differences in capabilities among such companies into the evaluation. The same consideration applies to the discussion on utilization rates.*

We appreciate the reviewer's valuable observation. To address this concern, we have carefully re-examined our data sources and revised the

manuscript accordingly. Two distinct issues arise: the availability of capacity data and of production data.

As shown in Figure 2 below (included in the Supplementary Material), the Global Iron and Steel Tracker provides excellent coverage for capacity—over 95 % globally and for most major producing countries, both developed and developing. In contrast, production coverage is substantially lower: Figure 3 shows that production data are incomplete, with coverage only about 54 % for the United States, even lower for China, and below 75 % for many other producers.

Because of this limitation, estimating plant-level capacity utilization directly from the GIST¹ dataset could introduce significant uncertainty. Moreover, as the reviewer rightly notes, **extrapolating utilization rates from plants for which production data is available to all plants in a given country may introduce bias** since data is usually available for large and more modern installations, whose utilization rates may be above the national average. To ensure consistency and representativeness, we therefore rely on country-level utilization rates for each technology, derived from World Steel Association (WSA) production data and GIST capacity data throughout the analysis. Figure 4 compares the country-level utilization rates for the BF-BOF steel production route computed with WSA production data with those estimated at the plant level with GIST production data. This graph shows significant bias in plant level utilization rates of developing countries.

A detailed discussion of these data issues has been added to the revised manuscript (see bottom of page 13 and top of page 14), Figure 4 has been added to the main paper and all other figures mentioned above have been added to the Supplementary Material. In addition, uncertainty associated with the projection of future utilization rates is illustrated in Figure 6 in the main paper and Figure 5 in Supplementary Material.

7. *Regarding cross-sector applicability, the methodology utilized in the steel sector is suggested to be applicable to other sectors. While it is evident that the broad framework can be applied, **the authors should demonstrate how the challenges in obtaining data for other sectors (such as cement, aluminum, pulp and paper, and power) differ from those encountered in the steel sector and how these differences may affect the results.** In addition to the unique challenges of applying this methodology across various sectors, summarizing the availability of plant-level data (e.g., plant closures and utilization rates) in a table would be beneficial. This summary could include the degree of technological progress and the adoption/diffusion of various energy efficiency and decarbonization technologies, which the authors have not fully considered thus far. Furthermore, providing concrete examples and data to*

¹GIST (Global Iron and Steel Tracker) is the new name (since 2025) of the Global Steel Plant Tracker database

support the claim of applicability to other sectors could lead to significant enhancements. It would also be helpful to indicate instances where evaluation processes distinct from those used in this steel study are required for specific sectors.

We thank the reviewer for this valuable comment. In response, we have introduced a new section (Section 2, Cross-sector applicability) in the Supplementary Material, which provides a detailed discussion of decarbonization pathways and the specific challenges of applying our methodology to four sectors: cement, aluminum, pulp and paper, and power. To address the request for a data summary, this section includes a table (Table 3, Plant-level and technology data availability for various sectors), which lists data sources and provides references to the relevant databases.

8. It is recommended to thoroughly evaluate the impact of uncertainty. As previously noted, future technological advancements and their rates of adoption/diffusion are critical factors. If it is challenging to examine these within the scope of this paper, it is essential to demonstrate that their impact on the results has been adequately assessed. Clearly defining the range of uncertainty and evaluating the sensitivity of the results to these uncertainties is recommended.

The different sources of uncertainty in our methodology are discussed in detail in Supplementary material, section “Sources of uncertainty in the asset level emission projection methodology”. This discussion was extended to include the uncertainty related to the future evolution of emission factors, as suggested by the reviewer.

In the revised version of the main paper, we include the estimates of uncertainty from two sources, considered to be most material: (i) uncertainty related to future utilization rates of plants, which was already evaluated in the initial submission; (ii) uncertainty related to future evolution of emission factors, following the reviewer’s suggestion.

To evaluate the latter uncertainty, we conduct the estimation procedure using the higher and lower bounds on future emission factors computed using the projection method discussed above in the answer to the second comment. In addition, by considering alternative decarbonization scenarios for the electricity sector, we capture the influence of this factor on overall uncertainty.

The main changes in the paper made to address this comment include the discussion of emission factor projections (pages 10 and 11 in the Methods section), new figures 2 and 3 in the main paper (emission factor uncertainty) as well as their discussion on pages 5 and 6, new figure 6 in the main paper and figure 5 in Supplementary material (utilization rate uncertainty).

9. *Additionally, it is crucial to assess whether the assumptions regarding the carbon intensity of future power generation are re-*

alistic, as these assumptions significantly impact the results. A careful examination of the basis for the scenarios and their consistency with historical data is necessary, along with sensitivity analyses on representative factors to validate the scenarios. Recent advancements in information technology have raised the possibility of a significant increase in future electricity demand, particularly drawing attention to the energy procurement strategies of IT companies. While there are indications that this demand may be met through decarbonized energy sources and nuclear power, there remains a divergence of opinions regarding how the balance of supply and demand will be forecasted across various scenarios. It would be beneficial to present these perspectives and incorporate sensitivity analyses for several scenarios to provide a more comprehensive understanding of the potential outcomes.

We fully agree with the reviewer that the future carbon intensity of electricity generation—and thus the emissions associated with steel production—is subject to considerable uncertainty. However, a detailed examination of the assumptions and internal consistency of power-sector decarbonization scenarios lies beyond the scope of this study. Our primary objective is to assess the alignment of the steel sector with specific, widely used prospective net-zero pathways—in particular, those developed by the International Energy Agency (IEA), which are the standard reference for alignment analyses [1].

Sectoral alignment analysis evaluates whether a given industry’s projected emissions trajectory is consistent with a predefined transition pathway, under the macroeconomic and energy-system assumptions of that scenario. In this sense, our claim is that steel-sector companies are not aligned with the IEA NZE scenario, meaning that even if the power sector decarbonizes at the rate assumed in this scenario, the steel industry would still fall short of its prescribed target. Evaluating consistency with other scenario frameworks, such as the Enerdata EnerOutlook scenarios (eneroutlook.enerdata.net/energy-scenarios-description.html) or the scenarios from the NGFS scenario database (ngfs.net/ngfs-scenarios-portal/), could be informative but would dilute the central message of our paper.

This being said, in the Supplementary Material (Section 2 and Figure 3), we compare the IEA scenarios for the decarbonization of the electricity sector with the corresponding scenarios from the NGFS scenario database, and find that for NZE and STEPS scenarios, the IEA decarbonization rates are more conservative than the NGFS ones and for the STEPS scenario, the IEA rate is within the range of rates obtained with different models for the corresponding NGFS scenario.

We also acknowledge that the evolving electricity consumption and procurement strategies of IT companies, particularly data centers could potentially alter the IEA projections and make these scenarios less relevant.

We have therefore performed a sensitivity analysis to evaluate the potential impact of this increase of consumption on the carbon intensity of electricity.

To this end, we refer to the IEA’s Energy and AI report (April 2025), which provides detailed projections of energy demand and supply for data centers through 2035. In its baseline scenario, global electricity use for data centers is expected to rise from 460 TWh in 2024 to around 1,000 TWh by 2030 (about 3% of global electricity generation). The largest increases are projected in mainland China (≈ 175 TWh, +170%), the United States (≈ 240 TWh, +130%), the European Union (≈ 45 TWh, +70%), and Japan (≈ 15 TWh, +80%). According to the report, about half of this additional demand will be met by renewable sources, with the remainder supplied by natural gas (20%), coal (20%), and nuclear power (10%). A regional breakdown of the supply mix is not provided.

To evaluate the implications, we estimated the updated projected emission factors for electricity in 2030 in the IEA STEPS and APS scenarios (because in NZE scenario there is no regional breakdown of electricity generation), applying the above assumptions together with default technology-specific life cycle emission factors from the IPCC Fifth Assessment Report.² Table 2 below compares these updated factors with the original baseline scenario values. The results indicate that the impact of additional generation linked to AI is relatively small and does not materially alter the main estimates of our paper.

This sensitivity analysis is now included in the Supplementary Material of the paper (Section 6 and Table 5).

10. *Discrepancies between the targets published by companies and the actual emission pathways may stem from data inconsistencies or methodological issues. To address this concern, verifying the transparency of the companies’ target-setting processes and reporting requirements is essential. For instance, in my experience, discrepancies have sometimes been noted between CDP data and the data published by individual companies. There may also be instances where the compilation year is outdated. Acknowledging these issues would be prudent.*

We appreciate the reviewer’s comment regarding the transparency of company target-setting processes. In the revised manuscript, we explicitly acknowledge that these processes may vary significantly across firms and that data inconsistencies can arise (see the paragraph on stated targets on page 12). Companies adopt different target-setting approaches—using different base years, absolute versus intensity targets, and varying degrees of disclosure.

Among the 36 companies in our database that have announced emission-reduction targets, only 5 have had their targets validated by the Science

²Coal: 820 gCO₂eq/KWh; Gas: 490 gCO₂eq/KWh; Renewable (Solar): 45 gCO₂eq/KWh; Nuclear: 12 gCO₂eq/KWh.

Based Targets initiative (SBTi), which offers some assurance of methodological consistency. While we recognize that targets from other companies may be less transparent or standardized, we believe that the substantial divergence we observe between the aggregate corporate target trajectory and our modeled decarbonization pathways cannot be explained solely by such data inconsistencies or methodological differences.

Comments from reviewer 2

*My only recommendation is to **clarify the nature of net-zero targets** (the difference and similarities between stated and IEA's targets). Are the net-zero targets/scenarios in IEA cumulative? To meet these goals then – should the emissions trajectories be cumulative? Currently, from my understanding of the presentation of the methods, it is not cumulative. It is only based on annual projections and that it was matched in terms of % reduction but not at the cumulative level. Could this be the reason why the gap is only around 20% rather than perhaps something like 50-70%? I could be wrong here, but maybe the authors can shed light on the nature of cumulative emissions net-zero targets in the IEA scenarios. Moreover, are the firm-level stated targets also cumulative or simply on a year-to-year annual production basis and how does that map to a cumulative emissions calculation vs non-cumulative.*

We appreciate the reviewer's comment and the opportunity to clarify this point. In both the IEA scenarios and our analysis, the net-zero targets are defined on an annual basis, not cumulatively. Accordingly, our comparisons are made between annual emission trajectories—that is, annual company- or sector-level targets—and the corresponding annual values from the IEA scenarios.

References

- [1] INSTITUT LOUIS BACHELIER, *The alignment cookbook - a technical review of methodologies assessing a portfolio's alignment with low-carbon trajectories or temperature goal.*, (2020).
- [2] B. TIKADAR, D. SWAMI, AND V. CHOWDHARY, *Process-level emission analysis and decarbonization pathway for bf-bof route in indian iron and steel industry*, *Journal of Environmental Management*, 373 (2025), p. 123483.
- [3] J. ZHANG, H. SHEN, Y. CHEN, J. MENG, J. LI, J. HE, P. GUO, R. DAI, Y. ZHANG, R. XU, ET AL., *Iron and steel industry emissions: a global analysis of trends and drivers*, *Environmental science & technology*, 57 (2023), pp. 16477–16488.

Figure 1: Statistical projection of BF-BOF emission factor for a sample of countries based on historical data.

Country-level capacity coverage in GSPT relative to OECD (2021)

Figure 2: Capacity coverage in GIST database. Production coverage in GIST database. The figure displays the ratio of total capacity listed in GIST database, to the total OECD capacity, for each country.

Country level capacity with production in GSPT relative to OECD (2021)

Figure 3: Production coverage in GIST database. The figure displays the ratio of capacity for which production data is available in GIST database, to the total OECD capacity, for each country.

Figure 4: **Comparison of country level BF-BOF utilization rates computed with World Steel Association data (dashed red lines) with plant-level BOF utilization rates computed with GIST data (blue bars).** Error bars correspond to one standard deviation of plant-level rates. Numbers on the bars correspond to the number of plants for which production data is available. For China, the GIST-based utilization rate in 2019 and 2020 is slightly higher than one because the actual production of some plants is higher than their declared capacity due to temporary capacity overload which is common in China. This graph shows important bias in plant level utilization rates due to the better production data availability for more modern plants, which are also likely to operate at higher utilization rates.

Process	Technology	CO₂ abatement (kgCO₂/tonne)
Coking	Coke dry quenching (CDQ)	50.05
	Coal moisture control (CMC)	16.32
Sinter	Heat recovery from sintering and sinter cooler	73.83
Blast Furnace	Pulverized coal injection	60.41
	Top-pressure recovery turbines (TRT)	27.3
	Recovery of BF gas	5.84
	Preheating of fuel and air for hot blast stove	26.03
BOF	Recovery of BOF gas and sensible heat	48.19
	Flue gas waste heat recovery	19.24
Casting	Continuous casting	36.35
Hot rolling	Recuperative or regenerative burner	46.5
	Process control in hot strip mill	25.61
	Waste heat recovery from cooling water	2.95
Cold Rolling	Heat recovery on the annealing line	30.06
	Automated monitoring and targeting systems	24.04

Table 1: Best available technologies for BF-BOF steel production route and their CO₂ abatement potential, based on [2].

Scenario	Country	Baseline EF	AI-adjusted EF
IEA STEPS	United States	164.00	165.95
IEA APS	United States	107.00	111.45
IEA STEPS	China	385.00	382.36
IEA APS	China	342.00	339.94
IEA STEPS	Europe	99.00	100.00
IEA APS	Europe	70.00	71.21
IEA STEPS	Japan	252.00	251.34
IEA APS	Japan	230.00	229.66

Table 2: Average emission factors for 2030 in IEA scenarios compared to emission factors adjusted for additional demand due to AI deployment.

Estimating firms' emissions from asset level data helps revealing (mis)alignment to net zero targets

Response to the reviews of the revised version

Hamada Saleh¹, Stefano Battiston^{2,3}, Irene Monasterolo^{4,5,6},
Thibaud Barreau¹, Peter Tankov^{7,1*}

¹Institut Louis Bachelier.

²University of Zurich.

³University of Venice.

⁴University of Utrecht.

⁵CEPR.

⁶WU Wien.

⁷CREST, ENSAE, Institut Polytechnique de Paris.

*Corresponding author(s). E-mail(s): peter.tankov@ensae.fr;

We thank both reviewers and the editor for their careful reading of the revised version of our paper. This document provides detailed answers to the reviewers' comments. A completed copy of the checklist with editorial revisions is uploaded separately.

Reviewer 1

I would like to suggest referencing previous works that are closely related to the methodology employed in this study. Specifically, there has been a study utilizing technology deployment rates (see below) that initiated similar approaches to estimating technology-based emission reduction potentials. Including such references could provide valuable context and further strengthen the scientific foundation of this section.

Our answer

The suggested reference was added to the reference list and quoted in the paper (see top of p.12)

Reviewer 2

My only recommendation and suggestion is that the authors be more upfront and clear about this key difference. The IPCC emissions budget, which is typically the ground reference for many carbon targets, is cumulative, but that the targets examined in this paper are not, and whether or not these annual targets translate to the appropriate cumulative carbon emissions targets we need to meet to remain below the 1.5/2 degree thresholds.

Our answer

To address this comment, the following paragraph was added in the results section (see bottom of p.7):

Note that the targets examined in this paper, as well as the IEA emission scenarios used for comparison, are formulated in terms of annual emissions, whereas IPCC carbon budgets are defined cumulatively. The use of annual emission targets is standard in the scenario literature, as it allows one to track the time profile of sectoral emission pathways and to analyze the dynamics of transition in the iron and steel sector.

Review of Estimating firms' emissions from asset level data helps revealing (mis)alignment to net zero targets

Summary

The authors develop a way to calculate the trajectories of firm-level emissions and benchmark this with net-zero targets. The authors combine four data sources, namely the IEA scenario data (for the targets), asset-level firm data, technology data (from the World Steel Association), and emissions data (from CDP and Refinitiv). The authors use projections of assets, its projected utilization and projected emissions factors to derive at the trajectory of (bottom-up) emissions paths. They then compare these trajectories with that of the EIA to determine whether firms are on-track or not in meeting net-zero targets. The authors focus on the iron and steel industry. They conclude that the iron and steel industry is projected to overshoot by 8-20% from the levels of the corresponding net-zero scenario.

Strengths of the paper

I commend the authors! They address a very important topic. The paper is well-written, well-documented, and very straightforward. I enjoyed reading the paper. My quick takeaway is actually a very optimistic one. The Iron and Steel industry is *only* about 20% away from their net-zero targets. This is actually very impressive. I would not have been surprised if it were about 50-60% overshoot from net-zero targets. Their calculation is already somewhat takes a conservative approach (applying highest emissions factor when there is ambiguity, picking the maximum reported scope between CDP and Refinitiv, etc.). I also applaud the author for using firm-level data from CDP and Refinitiv. These datasets are sometimes messy and frustrating to work with, so utilizing these data sources is an incredible task of its own.

The paper creates an important discussion not just for the iron and steel industry, but for all other industries and what the different challenges and opportunities are.

Opportunities for improvement

My only recommendation is to clarify the nature of net-zero targets (the difference and similarities between stated and IEA's targets). Are the net-zero targets/scenarios in IEA cumulative? To meet these goals then – should the emissions trajectories be cumulative? Currently, from my understanding of the presentation of the methods, it is not cumulative. It is only based on annual projections and that it was matched in terms of % reduction but not at the cumulative level. Could this be the reason why the gap is only around 20% rather than perhaps something like 50-70%? I could be wrong here, but maybe the authors can shed light on the nature of cumulative emissions net-zero targets in the IEA scenarios. Moreover, are the firm-level stated targets also cumulative or simply on a year-to-year annual production basis and how does that map to a cumulative emissions calculation vs non-cumulative.

Limitations

I have worked with the CDP data extensively. The CDP data is very challenging to work with. For Scope 1 location based data, sometimes you get missing data or a facility missing. But I don't think this completely negates the value of the study. It may just help to mention this nuance that sometimes firms' disclosure is not as comprehensive.

Recommendation

Overall, I am positive about the paper. I hope the authors take my comments/questions constructively as an innocent inquiry/clarification. I think shedding light on these variations on cumulative vs non-cumulative/annual production trajectories may strengthen the paper and make it more precise. It is important to compare apples to apples between IEA scenarios and these trajectories.

I wish the authors all the best!